# Construction of Non-Biaryl Atropisomeric Amide Scaffolds Bearing a C–N Axis via Enantioselective Catalysis

**DOI:** 10.3390/molecules27196583

**Published:** 2022-10-04

**Authors:** Xiao Xiao, Biao Chen, Yi-Ping Yao, Hai-Jie Zhou, Xu Wang, Neng-Zhong Wang, Fen-Er Chen

**Affiliations:** 1Institute of Pharmaceutical Science and Technology, Collaborative Innovation Center of Yangtze River Delta Region Green Pharmaceuticals, Zhejiang University of Technology, Hangzhou 310014, China; 2Research and Development Department, Zhejiang Hisoar Pharmaceutical Co., Ltd., Taizhou 318000, China; 3Hubei Key Laboratory of Natural Products Research and Development, College of Biological and Pharmaceutical Sciences, China Three Gorges University, Yichang 443002, China; 4Shanghai Engineering Center of Industrial Asymmetric Catalysis for Chiral Drugs, Shanghai 200433, China; 5Engineering Center of Catalysis and Synthesis for Chiral Molecules, Department of Chemistry, Fudan University, 220 Handan Road, Shanghai 200433, China

**Keywords:** non-biaryl atropisomers, C–N chiral axis, atroposelectivity, catalytic asymmetric synthesis

## Abstract

The significant scaffold offered by atropisomeric amides with a C–N chiral axis has been extensively utilized for pharmaceuticals, agricultural science, and organic syntheses. As a result, the field of atropisomer synthesis has attracted considerable interest within chemistry communities. To date, a range of catalytic atroposelective approaches has been reported for the efficient construction of these challenging scaffolds. However, greatly concise and highly useful methodologies for the synthesis of these atropisomeric compounds, focusing on transition-metal, chiral amine, and phosphoric acid catalysis reactions, etc., are still desirable. Hence, it is indispensable to succinctly and systematically present all such reports by means of disclosing the mechanistic analysis and application, as well as the challenges and issues associated with the establishment of these atropisomers. In this review, we summarize the development of catalytic asymmetric synthetic strategies to access non-biaryl atropisomers rotating around a C–N chiral axis, including the reaction methods, mechanism, late-stage transformations, and applications.

## 1. Introduction

Atropisomerism, a form of chirality caused by restricted rotation around a bond axis, has been of the utmost interest to chemists since its discovery in 1922 [1]. One particular category of non-biaryl atropisomeric amide scaffolds containing axial chirality along a C–N bond axis, comprising tertiary anilides, sulfonamides, sulfinamides, and phosphamide, has received more attention in the past few decades due to the discovery of their crucial roles in the pharmaceutical industry [2,3,4,5,6], agricultural science [7,8,9], and organic chemistry [10,11,12,13]. To date, the C–N axially chiral units have presented a series of significant natural products and bioactive molecules, which usually exhibit different pharmacological activities and metabolic processes, both in vivo and in vitro (Figure 1a) [4]. Telenzepine is a selective muscarinic antagonist assisting in the therapy of peptic ulcers [2]. When it comes to the field of agricultural science, *S*-metolachlor, a significant herbicide, features a C–N chiral axis structure [8,9]. Numerous efficient ligands and catalysts bearing C–N axially chiral backbones have also been synthesized and extensively utilized in organocatalysis and transition-metal-mediated asymmetric catalysis (Figure 1b) [10,11,12,13].

The conventional methods for preparing these optically active C–N axially chiral compounds generally relied on the optical resolution of the racemate, utilizing chiral column chromatography or chiral resolution agents [14,15]. Over the past few decades, our expanding knowledge of asymmetric catalysis has paved the way toward pioneering work on enantioselective synthesis. The catalytic synthesis of C–N axially chiral amides has now been established, via both asymmetric transition-metal catalysis and organocatalysis [16,17,18,19,20,21]. The atropisomeric amide scaffolds cover the acyclic and cyclic C–N axially chiral amides, including the anilides, sulfonamides, sulfinamides, phosphamides, hydrazides, succinimides, urazoles, aryl-2-pyridones, arylquinazolinones, etc. (Figure 1). As such, the structures that bear a C–N chiral axis without –(R)C=S, –(R)C=O, –(R)SO_x_, or –(R)_2_P=O bonds linking to the nitrogen atom will not be included. As elaborated at the beginning of this introduction, these frameworks have various utilities that highlight the necessity of discussing the enantioselective construction of these structures in a review.

In this mini-review, we aim to summarize the breakthroughs and recent advances in asymmetric catalyzed functionalization for the construction of C–N axially chiral amide scaffolds, enabled by both metal catalysis and organocatalysis strategies. We hope that this mini-review will highlight the great potential of these strategies in asymmetric syntheses and inspire further developments in this emerging field. For clarification, this mini-review is organized into two sections, according to the generation of different types of chiral products, including acyclic C–N, axially chiral amide, and cyclic C–N axially chiral amide. In the following part, the detailed performance of these scenarios will be depicted using specific examples.

## 2. Enantioselective Synthesis of Acyclic C–N Axially Chiral Amides

This section covers the most pertinent examples of synthesizing acyclic atropisomeric anilides, sulfonamides, sulfonamides, and phosphamides via transition-metal catalysis, phase-transfer catalysis, and nucleophilic catalysis.

### 2.1. Synthetic Strategies for Acyclic C–N Axially Chiral (Thio)anilides

#### 2.1.1. Atropisomeric (Thio)anilide Syntheseis via Transition-Metal Catalysis

The transition metal-catalyzed cross-coupling reaction has emerged as one of the most effective methods for the construction of C–C and C–Heteroatom bonds from readily available feedstocks and has also found extensive utilization in organic synthesis [22,23,24,25,26]. The booming demand for chiral compounds in the chemical and pharmaceutical industries has continued to facilitate the advancement of practical catalytic methods for asymmetric transformation. As a significant research area in asymmetric catalysis, the enantioselective transition-metal-catalyzed cross-coupling reaction has attracted increasing scientific attention, offering a general and direct technique for preparing enantiopure molecules [27,28,29]. In 2002, Kitagawa and co-workers generalized the first catalytic asymmetric *N*-alkylation strategy to afford atropisomeric anilide **3** from amide **1** and diallyl carbonate **2** (Figure 2) [30]. In this palladium-catalyzed alkylation, (*S*)-tol-BINAP (**L1**) served as the optimized ligand, to enable the efficient formation of the enantiopure axially chiral anilides **3** in high yields with low enantioselectivities (up to 96% yield and 44% *ee*). Although this method was inefficient for realizing the high enantioselective synthesis of atropisomeric amides by the limitation of ligands, it could illuminate the future of this cutting-edge area by providing novel and high-efficiency synthetic methodologies. Almost simultaneously, a similar method was then reported by Curran’s group, who achieved the catalytic asymmetric synthesis of axially chiral anilides with chiral palladium catalysts, up to about 50% *ee* [31].

On the basis of the earlier works by Kitagawa et al., Du and co-workers exploited a palladium catalytic system to gain access to axially chiral anilides via the Tsuji–Trost reaction (Figure 3) [32]. With the development of ligand design and formation, these asymmetric syntheses achieved dramatic improvements. Under the conditions of allylic substitutions, the process allowed the construction of atropisomeric amides with moderate enantioselectivity (up to 92% yield and 84% *ee*). The key to success in promoting enantioselectivity was the utilization of the P/olefin hybrid ligand **L2**. In contrast to the previous alkylation strategies, Du’s work would further improve the substrate scopes and enantioselectivities of the desired products.

Building on the previous study, Kitagawa’s group generalized a novel asymmetric arylation strategy to afford atropisomeric *N*-arylamides **9** (Figure 4) [33,34]. In this palladium-catalyzed arylation of amides **7** with aryl iodides **8**, 1-iodo-4-nitrobenzene served as the optimized substrate to enable an efficient formation of the enantiopure axially chiral anilides **9** in high yields, with excellent enantioselectivities (up to 84% yield and 94% *ee*). In contrast to the synthetic yields and enantioselectivities of **9a–9d**, the reactivity of the amides **7** might be rationalized on the basis of the nucleophilicity of the corresponding amide anions. That is, the yields of the desired products **9** were increased with the enhancement in the nucleophilicity of the related amide anions (the decrease in the acidity of NH-hydrogen).

The asymmetric [2 + 2 + 2] annulation is a highly effective protocol for the construction of axially chiral biaryls [35,36,37,38,39,40,41]. As illuminated by the previous works, a Rh(I)-catalyzed asymmetric enantioselective aromatization of trimethylsilylynamides **10** and 1,6-diynes **11** via [2 + 2 + 2] cycloaddition was reported independently by Tanaka and co-workers in 2006 (Figure 5) [42,43]. This method furnished enantioenriched axially chiral anilides with poor to good yields, with good to excellent enantioselectivity. It is worth mentioning that the preferential formation of **Int-1** determined the enantioselectivity of this cyclization, due to the coordination of the carbonyl group of **11** to rhodium and the steric interaction between the PAr_2_ group of (*S*)-xyl-BINAP and the benzyl group of **11**. The reductive elimination of rhodium species releases the rhodium catalyst and forms the desired product (*S*)-**12**. Notably, the carbonyl group on trimethylsilylynamides might serve a supporting role in gaining high *ee* values by its coordination with rhodium. The 1,6-diyne bearing a methoxycarbonyl group sharply decreased the enantioselectivity of the corresponding product, probably owing to the electronic repulsion between the carbonyl group of trimethylsilylynamides and 1,6-diyne (**12f**).

The above-mentioned enantioselective protocols were limited to conventional transition metal-catalyzed cross-coupling reactions. An asymmetric C–H functionalization has become one of the most efficient strategies in the formation of enantioselective structures [44,45,46,47,48,49]. As a significant research field in asymmetric catalysis, atroposelective C–H functionalization has attracted increasing attention, furnishing a novel technique for generating chiral molecules in a step- and atom-economical manner. The utilization of a C–H activation strategy by which to construct atropisomeric anilides was underdeveloped. To fill this gap, in 2020, Shi and co-workers described the first formal Pd(II)-catalyzed atroposelective C–H olefination to access a highly efficient synthesis of atropisomeric anilides **15** via using readily available *L*-pyroglutamic acid (**L5**) as the chiral ligand (Figure 6) [50]. This reaction underwent both an asymmetric C–H functionalization and a dynamic kinetic resolution process. Striving toward making this straightforward access a success, it should not only overcome the relatively complicated rotational freedom to fulfill the excellent enantioselective induction needed during the C–H activation step but also maintain the chirality that is forming, due to the relatively low atropostability of the resulting products, respectively. Notably, the utilization of the readily available and inexpensive ligand *(L*-*p*Glu-OH) to form the key intermediate **Int-2** and the development of the mild reaction conditions were essential for an elegant transformation. In terms of substrate scope, a wide range of anilides could be transferred to the anilide atropisomers, giving excellent yields and enantioselectivities (50 examples, with up to 99% yield and > 99% *ee*). Furthermore, a double C–H olefination of **14a** and a kinetic resolution of **14b** were completed with high efficiency. The scaffolds were then introduced to act as both an olefin-pyridine-type ligand (e.g., **15l**) with N–C axial chirality in the Rh-catalyzed asymmetric conjugate addition and a chiral Lewis base catalyst (e.g., **15j**) in the asymmetric allylation of aldehydes. Experimental studies were conducted to investigate the *N*–Ar rotational barriers (ΔG^‡^) and the negative correlation between *δ*_CO_ and the rotational barriers. These demonstrated that the atropostability of those atropisomeric anilides toward racemization depended on the electronic effects of both the aromatic ring and the picolinamide. Computational investigations were executed to reveal the mechanistic process and the asymmetric induction model of the atroposelective C–H olefination, leading to the brilliant access of synthetic applications by utilizing the axially chiral anilides to form bioactive molecules and develop asymmetric reactions.

Very recently, a similar Pd-catalyzed atroposelective C–H allylation with 1,1-di-substituted alkenes **23** was disclosed by the same group, leading to the formation of chiral *N*-aryl peptoid atropisomers **24** via *β*-H elimination, utilizing commercially available and inexpensive *L*-*p*Glu-OH **L5** as the chiral ligand (Figure 7) [51]. In this transformation, exclusive allylic selectivity was easily obtained, and numerous enantioenriched *N*-aryl peptoid products were generated in good yields, with high enantioselectivities (up to 90% yield and 97% *ee*).

In a similar vein, the concept of atroposelective C–H functionalization was focused on the construction of a wide variety of *N*-aryl peptoid atropisomers **27** in high yields, with excellent enantioselectivities as well (Figure 8) [52]. The Pd(II)-catalyzed asymmetric C–H alkynylation utilized the inexpensive *L*-pyroglutamic acid as a ligand. In this conversion, pyridine was indispensable; it served as the coordination site to accelerate the C–H activation rather than the carbonyl group or peptoid backbone. Moreover, the steric hindrance on the pyridine group would inhibit the coordination ability of palladium. Density functional theory (DFT) calculations identified that **Int-4**, leading to the formation of (*R*)-axial chirality, is more favorable than **Int-3**. When product **27b** was oxidized by *m*-CPBA, the corresponding pyridine *N*-oxide was furnished with the retention of enantioselectivity. These significant scaffolds could also be scalably formed and transformed in high fidelity into chiral aryl acetylene **27bb**, which is frequently applied in the Sonogashira cross-coupling and click reactions.

Kinetic resolution (KR), as one of the most traditional and fundamental approaches, has been extensively employed in asymmetric catalysis and synthesis, in which racemic substrates are resolved to furnish enantiomerically pure chiral complexes [53]. This general strategy was also concerned and applied to the synthesis of C–N axially chiral substances. In 2021, the Gong group developed a unique atroposelective C(sp^3^)–H coupling reaction and successfully achieved the kinetic resolution of thioanilides **28** (Figure 9) [54]. Under the catalysis of a hybrid palladium catalyst encompassing a phosphoramidite ligand **L6** and an anionic chiral cobalt complex **L7**, both atropisomeric arylated thioanilides **30** and *N*-methyl atropisomeric thioanilides (*R*)-**28** were readily obtained with excellent enantioselectivities via the kinetic resolution of atropisomeric thioanilides **28**.

#### 2.1.2. Atropisomeric Anilides Syntheses via Phase-Transfer Catalysis (PTC)

Phase-transfer catalysis (PTC) has long been identified as a versatile strategy for organic synthesis in both academic and industrial laboratories, characterizing its concise experimental operations, mild reaction conditions, environmentally benign and inexpensive reagents and solvents, and promising potential for accessing large-scale applications [55,56,57,58,59]. Hence, a broad avenue can be provided to obtain a highly selective methodology for the formation of enantioenriched axially chiral anilides.

In 2012, the first example of phase-transfer-catalyzed *N*-alkylations for the asymmetric synthesis of axially chiral *o*-iodoanilides **33** was reported independently by Maruoka and co-workers (Figure 10) [60]. Under the catalysis of chiral quaternary ammonium salt **Cat-1**, a highly enantioselective S_N_^2^ attack of anilides **31** to a range of bromides **32** was realized, leading to the generation of atropisomeric anilides at high yields, with moderate to excellent atroposelectivity. To verify the utility of these atropisomers, transformation operations were conducted. The axially chiral phosphine oxides **34** were obtained from *o*-iodoanilide **33a** with the full maintenance of stereoinformation, which acted as a dramatic forerunner for the design of novel chiral ligands and catalysts. The intramolecular radical cyclization of axially chiral *N*-allyl-*o*-iodoanilide **33d** was then carried out to form 3-methyl indoline **35**, with high chirality transfer from axial chirality to C-centered chirality. Furthermore, the XRD analysis from the structure of (*S*)-**36**, prepared from a chiral ammonium bromide, and *o*-iodoanilide **31a** was performed to obtain insight into the transition-state structure for the present asymmetric synthesis of axially chiral anilides. The information of crystal structure (*S*)-**36** indicated that the negative charge of the iodoanilide anion was delocalized, while the transition state **Int-6** was probably more favorable than **Int-7** for the prevention of steric repulsion between aryl groups (Ar) on phase-transfer catalyst and iodide on the anilide anion. The transition state, **Int-6**, could finally react with a bromide **32** to obtain axially chiral anilide **33** with the observed absolute configuration.

Building on the success in *N*-alkylation to form axially chiral *o*-iodoanilides **39**, Maruoka and coworkers then achieved the construction of axially chiral *o*-*tert*-butylanilides, possessing lower rotational energy barrier, with high yield and enantioselectivity by using piperidine-derived catalyst **Cat-2** (Figure 11) [61]. These fantastic scaffolds could be readily transformed to both sulfone products **40** through Michael addition and chiral alcohol **41** via the sequence of S_N_^2^ substitution and reduction. In the application process, the substrates’ enantioselectivity could be retained without erosion using steps.

#### 2.1.3. Atropisomeric Anilides Syntheses via Organocatalysis

Organocatalysis, designed to mimic biomimetic catalysis, is a powerful instrument by which to fulfill fascinating transformations [62,63,64,65,66,67,68,69]. To date, nucleophilic catalysis has experienced a range of developments regarding the construction of C–N axially chiral compounds by utilizing chiral amines [70,71,72,73,74] or isothioureas [75,76] as catalysts. The first exquisite amine-catalyzed asymmetric allylic alkylation (AAA) reaction to access C–N axially chiral scaffolds **45** was reported by Li and coworkers, which employed anilides **43** and Morita–Baylis–Hillman carbonates **44** as substrates, with the biscinchona alkaloid **Cat-3** as the catalyst (Figure 12) [70]. In this transformation, the axially chiral anilides were obtained in moderate to high yields with good enantioselectivities, stereoselectivities, and cis/trans selectivities. The stereochemical stability of the chiral anilides was proven to be regulated by electronics (**45a–c**), substitution pattern (**45c–d**), steric effects (**45e–f**), and solvents. These significant entities were easily transformed into the *α*-amino acid derivative **51** and indoline alkaloid scaffold **49**, with high chirality transfer.

### 2.2. Synthetic Strategies for Acyclic C–N Axially Chiral Sulfonamide and Sulfinamide

#### 2.2.1. Atropisomeric Sulfonamide Syntheses via Transition-Metal Catalysis

Recently, palladium-catalyzed allylation was also established to construct atropisomeric sulfonamide **52** in good yields, with the enantioselectivities satisfied (Figure 13) [77]. A chiral π-allyl Pd intermediate was formed in this approach; the highly asymmetric transformation was arduous as a result of the sulfonamide anion attacking the π-allyl carbon on the opposite site of the Pd atom (as shown in **Int-8**). The key to achieving the atroposelective *N*-allylation was the suitable basicity of the sulfonamide anion, which was incapable of abstracting the amide hydrogen of the ligand to form the inactive catalyst, **Cat-4**.

To date, the integrated methods for constructing highly diastereoselective amides containing an N–C chiral axis and vicinal stereogenic center have been extremely scarce. The challenge in the synthesis of those compounds would be increased by introducing the stereogenic center in an adjacent position, which enhanced the steric effect and decreased the substrate reactivity. The product possessed more than one chiral center, thereby also leading to the difficulty of achieving high diastereoselectivity. As a workaround for challenging the synthesis of these complicated scaffolds, the alkyloxyallenes acted at convenient synthons by which to access the novel amide entities bearing a N–C chiral axis and stereogenic center. Jiang et al. performed a direct atroposelective hydroamination of sulfonamides **55** with alkyloxyallenes **56**, utilizing Pd-catalysis to give the unique scaffolds of **57** in good yields, with excellent enantioselectivities and diastereoselectivities under mild reaction conditions (Figure 14) [78]. The electron-withdrawing and large steric hindrance groups in the sulfonyl unit provided the desired product with high diastereoselectivity. Besides this, the ortho substituents in the *N*-phenyl ring bearing bulky steric hindrance also led to accessing the atropisomers in terms of high *dr* values. The continued oxidation process successfully approached the axially chiral anilide analogs **58**, while the atropoisomeric iminium-ion-mediated conversion enhanced the synthetic potential of this method. Moreover, the *γ*-addition adduct **59b** from product **57** and TMSN_3_ could transfer to an eight-membered cyclic sulfonamide **60** with 78% *ee* for four steps.

#### 2.2.2. Synthetic Strategies for Acyclic C–N Axially Chiral Sulfonamide and Sulfinamide via Nucleophilic Catalysis

Recently, Zhao and coworkers disclosed an elegant asymmetric allylic alkylation reaction to approaching the chiral sulfonamide **62** bearing an allyl scaffold, by the utilization of sulfonamide **61** and Morita–Baylis–Hillman (MBH) carbonates **44** as substrates (Figure 15) [71]. This method showed excellent efficiency and enantiopurity in the formation of axially chiral *N*-aryl sulfonamides. In this note, the chiral compound, bearing a mono-ortho-*tert*-butyl substituent on the aryl group, possessed a low rotation barrier, while its enantiopurity decreased rapidly during the reaction process, which could be a significant guidance point for the chemists who were designing reactions. The desired product **62** was value scaffolds, which could easily be transformed into indoles **63**, **64**, and **65**, with the retention of enantiopurity. Furthermore, the catalytic kinetic resolution of the NOBIN analogs was achieved smoothly by this method. This led chemists to furnish various valuable chiral catalyst precursors.

Very recently, the Xiao and Chen groups reported a practical and concise organocatalytic atroposelective *N*-alkylation to efficiently approach sulfonamides bearing both an allene [79,80,81]. and an allyl entity, with high enantioselectivity (Figure 16) [72]. Significantly, this methodology also enabled selective N–H activation in the subsequent transformation toward functionalized sulfonamides and achieved the kinetic resolution of NOBIN analogs to obtain a series of chiral catalyst precursors. In this article, the racemization experiments demonstrated that substituted allenoate-sulfonamides presented higher rotational barriers than the related acrylate-sulfonamides. Notably, the universal synthetic transformation could be readily scaled up, which boded well for its wide applications in chiral molecular syntheses via ozonization and reduction.

In 2020, Dong disclosed a highly atroposelective *N*-acylation reaction of aniline-derived sulfonamides **72** with *α*,*β*-unsaturated carbonic anhydride **73**, using chiral isothiourea as the catalyst (Figure 17) [75]. This approach provided a facile and efficient avenue for the formation of atropoisomeric sulfonyl-substituted anilides in good yields, with high to excellent enantioselectivities. Two types of intermediates would be generated in this transformation process. The **Int-10** was preferred for constructing the chiral product (*R*)-**74** than the **Int-11** because of the presence of the bulk *o*-*tert*-butyl group on the phenyl ring, which affected not only sterical repulsion but also the π–π interaction between the substrates and the skeleton of the catalyst.

On a similar note, the isothiourea-catalyzed atroposelective *N*-acylation of sulfonamides by employing the *ent*-**Cat-6** was delineated by Zhao and co-workers (Figure 18) [76]. This method offered a facile and efficient process by which to access an array of atropoisomeric sulfonyl-substituted anilide products in good yields, with high to excellent enantioselectivities. It should be noted that the diminished size difference between the two ortho-substituents would inhibit the enantioselectivity (**77b** vs. **77a**). Moreover, these desired products possessed the potential to be aryl iodide catalysts.

Further results on cinchona alkaloid-based catalysis and the formation of complex C–N axially chiral and sulfoxide chiral compounds were reported by Li and coworkers, wherein they proposed the utilization of sulfinamides **80** and MBH carbonates **81** to achieve this transformation via an asymmetric allylic alkylation (Figure 19) [73]. Excellent enantioselectivity and suitability for high diastereoselectivity were obtained in this reaction. The results of stereochemical stability showed that the new atropoisomeric scaffold possessed a higher rotation barrier (**82a**) than the other reported axially chiral anilines’ rotation barriers (**83a**–**83c**) [16]. This demonstrated that the large steric hindrance of *tert*-butyl on the C–N axis was crucial for the stabilization effect.

### 2.3. Synthetic Strategies for Acyclic C–N Axially Chiral Phosphamide via Nucleophilic Catalysis

In a similar vein, Li extended chiral amine catalysis to the atroposelective *N*-allylic alkylation of phosphonamides **84** and MBH carbonates **44** to obtain axially chiral phosphamides **85** with hydroquinidine catalyst **Cat-7** (Figure 20) [74]. Apart from the sterically crowded diaryl phosphoryl unit, the bulkier *ortho*-substituents were essential for stereochemical stability (**85a–c**). The *ortho*-halogen group performed a pivotal role in terms of excellent stereocontrol (**85e**), in which the group would be predisposed to generate favorable noncovalent interactions with the hydrogen atoms in **Int-12**. In the transition state **Int-13**, the methyl group triggered steric repulsion instead.

When the diphenyl phenylphosphoramidate was employed in this transformation, only moderate yield and bad enantioselectivity were obtained (**85f**). It might be that the weak hydrogen bond interaction was difficult to initiate. A kinetic resolution process was easily achieved by utilizing a dissymmetric phosphamide **84g** as the substrate. As effective catalysts (**85g** and **85a**), these products were applied to the asymmetric catalysis. Notably, the iodine-bearing phosphoryl anilide **85a** was more practical than benzoyl and sulfonyl analogs, as well as the previously reported spiro chiral diiodide entity, in the enantioselective oxidative spirolactonization of phenolic compounds **88**.

### 2.4. Synthetic Strategies for Acyclic C–N Axially Chiral Hydrazide via Enantioselective Amination

The concept of making use of azodicarboxylates as an electrophilic amino source has been well established. This section will cover examples of the formation of acyclic C–N axially chiral hydrazides from azodicarboxylates.

#### 2.4.1. Atropisomeric Hydrazide Synthesis via Asymmetric C–H Amination

Jørgensen et al. pioneered the innovative cinchona–alkaloid-based organocatalytic atroposelective amination of naphthamides **90** and carboxamides **91** via the asymmetric Friedel–Crafts reaction (Figure 21) [10,11]. This process was conducted through an activated naphthoxide that generated a chiral ion pair with azodicarboxylate **91** by using a cinchona-alkaloid catalyst that subsequently transformed, so as to access naphthamides **92** with excellent stereo- and regioselectivity. Significantly, the cinchona-alkaloid catalysts, **Cat-8**, themselves occurred with the Friedel–Crafts reaction, leading to the formation of a new class of cinchona-alkaloids **Cat-9** and **Cat-10** that promoted the atroposelective amination more efficiently, generating product **92** with high enantioselectivities and enabling an approach to both enantiomers of the products. The modified cinchona-alkaloid catalyst also successfully achieved an asymmetric Michael addition. Interestingly, the axially chiral catalyst **92a** was smoothly employed in the high-efficiency fluorination, giving rise to the desired product **95**, with high optical purity.

Chiral phosphoric acids (CPAs) have been widely employed in atropisomer synthesis [17,19,82,83]. In this context, Zhang and coworkers reported the CPA-catalyzed enantioselective C–H amination of *N*-aryl-2-naphthylamines **96**, with azodicarboxylates **97**, which directly furnished a broad range of substituted C–N atropisomers **98** with excellent enantioselectivity (Figure 22) [84]. This method could easily be applied to achieve the late-stage modification of the estrone analog. The stereo-stability study demonstrated that the intramolecular H-bonding interaction of the carbonyl group with the N-H bond was pivotal to the stereo-stability of these C–N axially chiral products. The putative reaction pathway was commenced on a simultaneous dual H-bonding activation from a self-assembly combination of CPA **Cat-11**, *N*-phenyl-2-naphthylamine **96** and azodicarboxylate **97**, leading to the formation of intermediate **Int-15**. The nucleophilic addition of *N*-phenyl-2-naphthylamine to the azodicarboxylate then occurred, resulting in the generation of intermediate **Int-16**. An efficient central-to-axial chirality conversion was easily realized via rearomatization, followed by the generation of product **98**. Notably, the π-π interaction in **Int-15** and **Int-16** was essential for the enantioselectivity control.

#### 2.4.2. Atropisomeric Hydrazide Synthesis via an Au-Catalyzed Cycloisomerization–Amination Cascade Reaction

Another characteristic example was reported by Gong and coworkers and showcased chiral gold(I) complex catalysis for enabling the enantioselective cycloisomerization–amination of 2-(alkynyl)phenyl boronic acids **99** and azodicarboxylates **100**, to furnish the atropisomeric hydrazide **101** (Figure 23) [85]. A wide scope of substrates possessing various functional groups was tolerated to form axially chiral hydrazide derivatives in high yields and fairly good enantioselectivities. The probable reaction pathway was initiated from the coordination of Au(I) complex to the C-C triple bond of 2-(alkynyl)phenyl boronic acids **99**, leading to the formation of vinyl-gold intermediate **Int-17**. A stereoselective attack of **Int-17** to gold-activated azodicarboxylates subsequently occurred to generate the species **Int-18**, followed by a proton transfer to obtain the desired product **101** and release the Au(I) complex for the next catalytic cycle.

## 3. Enantioselective Synthesis of Cyclic C–N Axially Chiral Amides

This section covers the most pertinent examples of synthesizing acyclic C–N axially chiral amides from prochiral precursors, such as the arylmaleimides, atropisomeric urazoles, spirobenzazepinones, arylpyridones, arylquinazolinones, and the other types of atropisomeric anilides.

### 3.1. Synthetic Strategies for C–N Axially Chiral N-Aryl Succinimides via the Desymmetrization of N-Arylmaleimides

The dominant method to assemble cyclic C–N axially atropisomeric amides is generalized by desymmetrization of *N*-arylmaleimides and triazodiones, atroposelective *N*-annulation, and atroposelective functionalization on preformed nonbiaryl C–N scaffolds. In this part, the catalytic asymmetric desymmetrization of N-aryl prochiral precursors requires a remote stereochemical control; undoubtedly, it is more difficult to enable excellent stereochemical controls of two chiral units via the desymmetrization process.

#### 3.1.1. Synthetic Strategies for C–N Axially Chiral *N*-Aryl Succinimides via Transition-metal Catalyzed Desymmetrization of *N*-Arylmaleimides

The first rhodium-catalyzed desymmetrization of *N*-arylmaleimides was reported by Hayashi and coworkers, wherein the asymmetric 1,4-addition reaction of phenylboronic acids **29** to maleimides **102** was employed to form atroposelective *N*-aryl succinimides **103** with high enantioselectivity and diastereoselectivity (Figure 24) [86]. Moreover, the desired product **103a** could be smoothly transformed to the complex **104**, bearing a quaternary carbon stereocenter via S_N_2 substitution, leading to the access of a chiral substituted succinic acid **105**. In addition, a subsequence of oxidation and Diels–Alder reaction easily followed, giving a chiral cycloadduct **107** in a high diastereomeric ratio.

Subsequently, Feng and co-workers disclosed an elegant catalytic enantioselective Michael addition/desymmetrization reaction of *N*-(2-*t*-butylphenyl)maleimides **109** with unprotected 3-substituted-2-oxindoles **108**, creating oxindole-succinimides **110** containing a C–N axis in good yields with good to excellent diastereo- and enantio-selectivities by using a chiral *N*,*N*’-dioxide-Sc(III) complex as catalyst (Figure 25) [87]. In this transformation, the various functionalized oxindoles **108** were well tolerated, while the scope of *N*-aryl prochiral precursors **109** was limited.

In 2016, Wang et al. further developed a highly efficient Ag(I)-catalyzed atroposelective desymmetrization of *N*-(2-*t*-butylphenyl)maleimide **109**, through the 1,3-dipolar cycloaddition of in situ-generated azomethine ylides from readily available imino esters **111**, leading to a facile generation of numerous biologically important and enantioenriched octahydropyrrolo [3,4-c]pyrrole derivatives **112** in high yields and with excellent levels of diastereo-/enantio-selectivities (Figure 26) [88]. The further oxidative transformations were successfully achieved by utilizing DDQ as oxidant, leading to the formation of fascinating 2*H*-pyrrole **113a** and poly-substituted pyrrole **113b**, respectively.

Recently, Xu and coworkers developed the palladium-catalyzed atroposelective hydrosilylation of *N*-maleimides **114** with hydrosilanes **115**, via remote control of the axial chirality strategy, giving a series of silyl succinimides **3** bearing a C–N axis in good yields, with good diastereo- and enantio-selectivities (Figure 27) [89]. The key to success with the perfect and remote control of axial chirality lay in the influence of bulky substitutions on the aryl ring. Moreover, this process possessed high functional group compatibilities.

More recently, Li and coworkers successfully created *N*-aryl succinimides **119**, containing an axially chiral axis and a central chirality via a single stereo-determining step, in which the rhodium-catalyzed C–H alkylation of benzamides **117** was utilized, with *N*-arylmaleimides **118** as the alkylating reagent (Figure 28) [90]. This transformation featured mild reaction conditions, broad functional group tolerance, and excellent enantio- and diastereo-selectivity. The key to the installment of the distally disposed axial and central chirality lay in the judicious choice of the chiral rhodium cyclopentadienyl catalysts. which enabled researchers to control both the orientation of the olefin element and the prochiral C–N bond.

Subsequently, Cheng, Fang and co-workers achieved the nickel-catalyzed enantioselective hydrocyanation of *N*-aryl 5-norbornene-endocis-2,3-dicarboximides **120**, leading to the fabrication of *N*-aryl succinimides **121**, bearing both five contiguous stereogenic carbon centers and one remote C–N axial axis, with excellent enantioselectivities (Figure 29) [91]. Mechanism studies indicated that the rigid structure of the cyclic imide was crucial for stereoselective control, and the existence of the imide carbonyl group was necessary for this transformation.

#### 3.1.2. Synthetic Strategies for C–N Axially Chiral N-Aryl Succinimides via the Organo-Catalyzed Desymmetrization of N-Arylmaleimides

To date, hindered imides, acting as prochiral precursors, are widely applied in construction of C–N atropisomers. The representative works were disclosed by Bencivenni’s group, in which a series of organocatalytic nucleophilic desymmetrization reactions of *N*-(2-*tert*-butylphenyl)-maleimides were well developed. In this case, the hindered rotation of bulky *^t^*Bu group on the aryl ring achieved this atropostability. Mechanistically, the *Re* or *Si* atropotopic face of maleimide substrates could be recognized by an organocatalyst to conduct a direct facial selective nucleophilic attack, away from the steric hindrance on the favorable side away from the *^t^*Bu group, leading to the construction of the distal C–N axis stereogenic and concomitantly forming stereocenters (Figure 30) [92,93,94,95,96].

The first organocatalyzed desymmetrization of *N*-(2-*t*-butylphenyl)succinimides **123** to furnish remote C–N atropisomers **124**/**125** was reported by Bencivenni and coworkers via the vinylogous Michael addition of 3-substituted cyclohexenones **122** to *N*-arylmaleimides **123**, in 2014 (Figure 31) [93]. In this transformation, the combination of 9-amino(9-deoxy)epi-quinine **Cat-12** and the acid, **Cat-13**, facilitated enantioselective desymmetrization, leading to the access of atropisomeric succinimides with two adjacent stereocenters in the requisite enantioselectivities. Regrettably, the catalyst system ineffectively controlled the diastereoselectivities. When the cooperative catalyst system was loaded to separate the solutions of pure **124a** and **125a**; after 24 h, the same 70:30 ratio of **124a** and **125a** in both mixtures was found, which demonstrated that catalyst-promoted dynamic epimerization was occurring at the exocyclic stereocenter, allowing the assignment of the ratio of epimers **124a** and **125a**. As shown in **Int-22**, the configuration of **124f** might be formed by the *Si*-face-selective nucleophilic attack.

Another desymmetrization example of the organocatalytic atroposelective formal Diels–Alder reaction with 3-substituted cyclohexanones **109** and enones **126** was realized with the identical 9-amino(9-deoxy)-epi-quinine catalyst **Cat-12** to access new succinimide atropisomer scaffolds **127** containing a chiral axis and three stereogenic centers (Figure 32) [94]. Except for *ortho*-bromo-substituted phenyl, alkyl, or ester enones, enones bearing both electronically differentiated phenyl and thienyl moieties obtained *endo*-diastereomers under excellent control of atroposelectivities and diastereoselectivities. In the more favored transition state, the H-bond interaction with the carbonyl group of maleimide was produced. Subsequently, the *tert*-butyl group crucially controlling an enamine addition from the less-crowded bottom plane led to access the desired products in **127**. Interestingly, inversed configurations at the three stereogenic centers resulted when only the *tert*-butyl group was absent. Notably, the scaffolds ***ent*-127** were smoothly afforded by utilizing the **Cat-14** as an organocatalyst. Moreover, this cinchonidine-catalyzed desymmetrization strategy was further utilized for the Michael additions of active carbon nucleophiles, such as *α*-acetylcyclopentanones, *α*-acylbutyrolactones, 2-cyano-2-phenylacetates, and oxindoles [95,96]

The *N*-heterocyclic carbene (NHC)-catalyzed transformation has emerged as a powerful strategy for the construction of significant scaffolds in the field of organic synthesis [97,98,99,100,101,102]. Recently, Biju and coworkers reported an NHC-catalyzed atroposelective desymmetrization of prochiral *N*-aryl maleimides **123** to access C–N axially chiral compounds **129** bearing a spiral ring skeleton (Figure 33) [103]. This elegant process underwent an intermolecular Stetter-aldol cascade of dialdehydes **128** with *N*-aryl maleimides **123** followed by one-pot oxidation, leading to approach the C–N axially chiral *N*-aryl succinimides **129** in good yields with excellent *ee* values. This transformation featured remote axial chirality control, broad scope, and mild conditions. The proposed mechanism commenced on the generation of a free carbene **Int-24** from the chiral triazolium salt **Cat-16**. A nucleophilic addition of the dialdehyde **128a** and the subsequent intramolecular proton transfer would smoothly afford the Breslow intermediate **Int-26**. The *Re*-face nucleophilic addition of species **Int-26** to the electron-poor double bond of the maleimide **123a** generated the enolate intermediate **Int-27**, followed by a proton transfer as the enantio-determining step to obtain the amide enolate **Int-28**. The intermediate **Int-28** could be converted to carbinol **Int-29** in two routes. In the first scenario, the intramolecular aldol process took place before the dissociation of NHC, leading to the generation of the complex **Int-29**. In the alternative pathway II, the aldol reaction occurred in the absence of NHC, and **Int-30** was subsequently formed, followed by a final proton transfer to afford the carbinol **Int-29**. The DFT computational calculations revealed that the aldol process was likely to occur after the dissociation of NHC. Ultimately, the oxidation of carbinol **Int-29** could furnish the desired product **129a** by utilizing PDC as oxidant.

#### 3.1.3. Synthetic Strategies for C–N Axially Chiral N-Aryl Succinimides through Miscellaneous Strategies

In 2009, Tan and coworkers disclosed a conjugate addition/enantioselective protonation cascade reaction of *N*-substituted itaconimides **130** with thiols to obtain the axially chiral *N*-phenyl-itaconimides by using chiral bicyclic guanidine **Cat-17** as catalyst (Figure 34) [104]. In this transformation, two atropisomers **131** and **132** were formed in high yields. These compounds possessed high energy barriers, providing a potential means for onward chirality transfer to some charming scaffolds.

In yet another report, Li et al. disclosed a brand-new palladium-catalyzed atroposelective construction of amides **134** with high efficiency through the carbonylative process (Figure 35) [105]. By interacting with aryl iodides **133** and carbon monoxide (CO), a series of cyclic axially chiral amides **134** were furnished in good yields, offering excellent enantioselectivities. Notably, this reaction can be employed to synthesize the drug candidates for inhibiting NF-κB activation in HeLa cells.

### 3.2. Synthetic Strategies for C–N Axially Chiral Urazoles via Desymmetrization

The *N*-arylmaleimide analogs, 4-aryl-1,2,4-triazole-3,5-diones (ATADs), have been widely applied as part of an asymmetric desymmetrization strategy for the establishment of C–N axial chirality. Recently, the Tan group developed a desymmetrization reaction to achieve the synthesis of urazole-type molecules encompassing axial chirality through the organocatalytic asymmetric tyrosine click-like addition of triazolediones **135** (Figure 36) [12]. The reaction was very compatible with the 2-substituted indoles **136** and 2-naphthols **137** by the utilization of the catalysis of chiral phosphoric acid (CPA) **Cat-18** and bifunctional thiourea-tertiary amine catalyst **Cat-19**, respectively. Excellent remote axial chirality control resulted from the efficient recognition of the two reactive sites in triazoledione, and the transfer of stereochemical information into the prochiral axis, far from the site of addition, was realized.

Based on the electrophilicity and a low N=N π-bond dissociation energy of prochiral ATADs, a three-component ene reaction of substrates **135** with in situ-generated stereodefined spirooxindoles **143** from the bisthiourea-catalyzed Diels–Alder reaction between methyleneindolinones **140** and 3-vinyl-indoles **141** was efficiently developed to approach spirooxindole–urazoles **142** in high yields with configurationally defined with the C–N axis and carbon centers (Figure 37) [106]. In the Diels–Alder reaction, the spatial configuration of intermediates **143** was determined by the bifunctional bisthiourea catalyst, through the proposed intermediate **Int-31**. Moreover, the aromaticity-driven ene reaction, proceeded with substrate-controlled enantioselectivity and led to forming the chiral spirooxindole–urazole **142**, in which ATAD access to the alkene from the opposite face of sterically hindered ester group and the bicyclic moiety was the favored process.

Very recently, the Chi group reported an NHC-catalyzed desymmetrization of a prochiral urazoles **145** for an atroposelective construction of the urazole **146** bearing a chiral C–N axis (Figure 38) [107]. An Enantioselective Michael addition of a nitrogen atom on the *N*-arylurazole **145** to an ynal-derived acetylenic acylazolium species **Int-32** was the pivotal process and yielded the intermediate **Int-33**, thereby the C–N axially chiral *N*-arylurazole **146** was furnished in excellent yields and with high *ee* values. Furthermore, these useful scaffolds could not only be obtained in gram scales but also readily achieve the synthetic applications, such as a sequence of reduction and amidation to access the urea **148e**.

### 3.3. Synthetic Strategies to Access Atropisomeric (iso-)Indolinone

Isoindolinones are significant pharmacophores, and they also extensively exist in various natural products. Nevertheless, catalytic atroposelective formation of *N*-aryl-isoindolinones bearing a C–N axial chirality remained to be unexplored until very recently. In 2017, Seidel et al. disclosed a catalytic atroposelective synthesis of isoindolinones **151** (Figure 39) [108]. A highly efficient enantioselective biomimetic transformation between 2-acylbenzaldehydes **149** and anilines **150** gave rise to approach C–N axially chiral *N*-aryl-isoindolinones **151** containing a central chirality with the help of CPA **Cat-22**. The catalytic reaction occurred by the generation of the cyclic bis-hemiaminal **Int-34** in the presence of CPA, followed by a sequence of dehydration and tautomerization, leading to the formation of active 2*H*-isoindole **Int-35**. A subsequent stereoselective-control tautomerization proceeded to give the desired products **151**. Moreover, the methodology was successfully utilized for creating the first generation of mariline A.

The C–H activation of arenes to fabricate C–N axially chiral elements represents a promising approach and remains highly desired and sought after. In 2019, Wang and co-workers developed the first example of employing C–H functionalization strategy to access a variety of C–N axially chiral *N*-aryloxindoles **154** in high yields with excellent enantioselectivities, in which an asymmetric Satoh–Miura-type process involving dual C–H activation proceeded (Figure 40) [109]. Mechanistically, the reaction began with oxidization to access CpRh^III^ from the chiral CpRh^I^ by using AgNTf_2_ as the oxidant. The subsequent coordination of the oxygen atom of **152a** to the Rh^III^ catalyst promoting the first C–H bond activation occurred through a concerted metalation-deprotonation (CMD) mechanism, leading to the formation of the six-membered rhodacyclic intermediate **Int-36**. The subsequent insertion of aryl alkyne **153** to **Int-36** occurred, resulting in the formation of the complex **Int-37**. Subsequently, the cleavage of the second C–H bond successfully managed to yield the species **Int-38**. Another equivalent of **153** inserted into the intermediate **Int-38** then generated the intermediate **Int-39** or **Int-39′**, followed by reductive elimination to furnish the product **154a** and release the CpRh^I^ species. Ultimately, to close the catalytic cycle, the real catalyst, CpRh^III^, was regenerated from CpRh^I^ with the assistance of the oxidant (Ag_2_O). Preliminary mechanistic studies figured out that the C–H cleavage step was not the turnover-determining step.

### 3.4. Synthetic Strategies to Access Cyclic C–N Axially Chiral N-Aryl Piperidinone/Pyridones/Quinolinone/Phenanthridinone

#### 3.4.1. Construction of Cyclic C–N Axially Chiral N-Aryl-Piperidinone via Brønsted Base Catalysis

Azacycles and oxacycles are significant scaffolds that are widely employed in numerous natural products and valuable fine chemicals. In 2010, Tan and coworkers disclosed an efficient process to access oxacycles and azacycles through a Brønsted base-catalyzed tandem alkyne [110] isomerization–Michael reaction sequence (Figure 41) [111]. In this transformation, one of the modified alkynyl amides **155** was employed inan intramolecular aza-Michael reaction on an in situ-generated allene (**Int-40**) via chiral guanidine catalysis, leading them to approach an axially chiral lactam with high enantioselectivity.

#### 3.4.2. Construction of Cyclic C–N Axially Chiral *N*-Aryl-Piperidinone via Brønsted Base Catalysis

The scaffold of 2-pyridone is a significant heterocycle, one that is more commonly found in pharmaceuticals and agrochemicals [112,113]. In this context, Tanaka and coworkers realized the enantioselective synthesis of the *N*-aryl-2-pyridone **158**, bearing a C–N chiral axis through cationic rhodium(I)–catalyzed asymmetric [2 + 2 + 2] annulation reaction of the 2-substituted phenyl isocyanate **157** with the alkyne **11** (Figure 42) [114]. A probable mechanism for the rhodium-catalyzed atroposelective [2 + 2 + 2] cycloaddition began with the cyclometallation of isocyanate **157** and alkyne **11** with rhodium, leading to accessing the intermediate **Int-41** or **Int-42**, depending on the structure of the substrates. The cycloaddition of less-coordinating diynes and/or more-coordinating isocyanates with rhodium showed a preference to form the intermediate **Int-43**. On the contrary, the reactions of more coordinating diynes and/or less coordinating isocyanates with rhodium led to the generation of species **Int-42**. Due to the steric effect between the aryl group of **11** and the chiral ligand, the formation of **Int-41** gave rise to higher enantioselectivity than that of **Int-42**. Insertion of alkyne or isocyanate would obtain intermediate **Int-43** and the subsequent reductive elimination of rhodium formed the atropisomeric pyridone **158**.

Later on, a similar strategy for the [2 + 2 + 2] cycloaddition was also successfully achieved by the Takeuchi group, via the catalytic system of [Ir(cod)Cl]_2_/BINAP, in which α,ω-diyne **159**, cyclized with the isocyanate **160** to give the C–N axially chiral 2-pyridone **161** in high efficiency. Both aliphatic and aromatic isocyanates were well tolerant of this transformation (Figure 43) [115]. Moreover, aromatic isocyanates were less reactive than aliphatic isocyanates due to its electronic properties. Mechanistically, an iridium active species reacted with the diyne **159** to obtain the iridacyclopentadiene **Int-44** via an oxidative cyclization. An isocyanate **160** then interacted with iridacyclopentadiene **Int-44** to form **Int-45**, and finally transformed it to the 2-pyridone **161**.

#### 3.4.3. Atropisomeric Quinolinone/Phenanthridinone Syntheses via Transition Metal Catalysis

In the above-mentioned strategy of synthesizing acyclic atropisomeric anilides, Kitagawa et al. also applied this Pd-catalyzed intramolecular atroposelective *N*-arylation to construct the cyclic atropisomeric products **163** from substituted amides **162**, with high enantioselectivities (Figure 44) [33,34]. This pioneering work provided a promising orientation by which to access atropisomeric arylheterocycles.

Shortly after, Kitagawa et al. reported another perfect example of the palladium-catalyzed enantioselective synthesis of C–N axially chiral phenanthridin-6-one derivatives **165** bearing numerous *ortho*-substituted phenyl groups on the nitrogen atom through the palladium-catalyzed intramolecular Buchwald–Hartwig amination from the amide **164** (Figure 45) [116]. The enantioselectivity of this transformation was extremely reliant on bases, solvents, and reaction temperature, as well as on the steric hindrance of *ortho*-substituents.

Copper-catalyzed enantioselective coupling reactions have attracted considerable attention from the chemistry communities over the past few years. In particular, these coupling strategies have become a very powerful tool for the construction of C–C, C–N, C–O, and other carbon–hetero-atom bonds, as well as for the synthesis of heteroatom-containing ring systems [117,118,119,120,121]. Whereas there are rare reports making use of copper-catalyzed asymmetric coupling reactions for the formation of atropisomers, in 2019, Gu and coworkers disclosed a novel Cu/N,N-(cyclohexane 1,2-diyl) dipicolinamide catalyst system, in which an intramolecular atroposelective Ullmann-type amination was successfully achieved for the generation of C–N atropisomers **167** from acyclic amides **166** (Figure 46) [122]. This process could obtain the desired compounds **166** in high yields, with excellent enantioselectivities. Notably, selected C(sp^2^)-Br bond cleavage and Gram-scale operations were easily achieved, providing access to bromo-substituted products **167g** and **167h** with high efficiency. In addition, further transformation was readily applied to form the potential chiral phosphine ligand **170**. The catalytic process commenced with the formation of complex **Int-47** in the presence of NaOH, the copper catalyst **Cat-23**, and substance **166**. A subsequent intramolecular oxidative addition of **Int-47** furnished the Cu(III) intermediate **Int-48** or **Int-49**. Alternatively, the directed oxidative addition of Cu(I) **Cat-23** to the bromide **166** perhaps underwent the generation of Cu(III) species **Int-50**, followed by a ligand substitution from bromide to amide to form **Int-48** or **Int-49**. Less steric repulsion was produced in **Int-48** because of the picolinamide group located in the downward position. Conversely, the ortho group in the aniline moiety of **Int-49** possessed a strongly steric repulsion with the upward picolinamide unit. Ultimately, the reductive elimination of **Int-48** constructed a new axial C–N bond to obtain the atropisomer **167** with an *R* absolute configuration.

Very recently, Zhou and coworkers disclosed a highly efficient synthetic strategy to access C–N axially chiral phenanthridin-6-one derivatives **173** from widely available substrates **171** and **172** (Figure 47) [123]. In this process, an exquisite palladium/chirla NBE* cooperative catalysis (Catellani reaction) was employed [124]. Logically, a sequence of C–H activation, oxidative addition, and reductive elimination proceeded to yield the key axially chiral palladium(II) species **Int-51**, followed by intramolecular amidation (**Int-52**) and the crucial axial-to-axial chirality transfer, to eventually produce C–N axially chiral phenanthridinones **173** with high fidelity. The proposed mechanistic routes were confirmed by the DFT’s computational calculations. Interestingly, further transformation studies were realized, giving access to the medium-sized lactone **174**.

In yet another example from He and coworkers, the first atroposelective establishment of C–N axially chiral enamides **177** from substituted cinnamyl carbonates **175** and 2-quinolinols **176** was disclosed via Ir-catalyzed asymmetric allylic substitution-isomerization, in which elegant hydrogen-bonding promoted central-to-axial chirality transfer. This was easy to achieve and provided entry to a great variety of quinolone- and pyridone-derived axially chiral enamides **177** in high yields, with excellent enantioselectivities (Figure 48) [125]. Mechanistically, substrates **175** and **176** underwent an Ir-catalyzed asymmetric allylic amination to obtain the preponderant products **178**, followed by H-bonding interaction with 1,8-diazabicyclo[5,4,0]-7-undecene (DBU) to form the complex **Int-53**. Subsequent isomerization occurred via a hydrogen bonding interaction and promoted [1,3]-H transfer [126], giving rise to the enantioselective formation of the desired enamides **177** with high fidelity. Significantly, hydrogen bonding interaction was essential for reactivity and stereospecificity during central-to-axial chirality transfer, which not only played a key role in the stabilization of the deprotonation transition state and the chiral ion-pair species but also suppressed the intramolecular steric repulsion of the intermediate stage, to greatly inhibit the generation of the undesired enantiomer. Furthermore, a valuable approach was employed to synthesize the chiral epoxide derivatives through the epoxidation of axially chiral enamides.

#### 3.4.4. Synthetic Strategies to Access Cyclic C–N Axially Chiral *N*-Aryl-Quinazolinones/Thiazines

Whereas the catalytic asymmetric desymmetrization of *N*-arylmaleimides was widely applied to atroposelectively construct atropisomers bearing C–N axes, there are scarce reports that other prochiral axial compounds are employed as substrates. In 2016, Kitagawa and coworkers developed the palladium-catalyzed reductive enantioselective desymmetrization of quinazolinones **179** for direct access to axially chiral mebroqualone derivatives **180** (Figure 49) [127]. The reductive efficiency of this enantioselective mono-hydrodebromination was not particularly high, due to the formation of byproducts **181** via over-reduction. Moreover, a Pd-catalyzed transformation was readily achieved to obtain the bioactive methaqualone in low *ee*, which was the GABA receptor agonist.

By taking advantage of the de novo construction strategy, Tan and coworkers reported the first catalytic atroposelective formation of C–N axially chiral *N*-arylquinazolinones **186** (Figure 50) [128]. This novel one-pot transformation was commenced on a CPA-promoted hemiaminal formation, followed by an oxidative dehydrogenation to furnish numerous aryl-quinazolinones bearing a C–N axis. When 4-methoxypentenones **185** were utilized as condensation partners, and more acidic *N*-triflylphosphoramide **Cat-25** was employed as catalyst, a carbon–carbon bond cleavage strategy was also realized to synthesize the axially chiral arylquinazolinones. Significantly, the practicability of this methodology was presented in the easy total atroposelective synthesis of the natural product, eupolyphagin, in high enantioselectivity.

Enantioselective aryl functionalization of C–N-containing prochiral synthons is a significant strategy to the atroposelective construction of an axially chiral C–N bond. However, the key to success in the elegant reaction process lies in: (1) taking full advantage of existing structural scaffolds; (2) effective creation of the structural complexity of the desired products; (3) introduction of the C–N axial chirality in high efficiency. In 2015, Miller et al. developed a pioneering investigation on the asymmetric construction of 3-arylquinazolin-4(*3H*)-ones **193** via an atroposelective bromination process (Figure 51) [129,130].

With the assistance of the tertiary amine-substituted *β*-turn peptide catalyst **Cat-26**, the tribromination of quinazolinones **192** bearing a pre-existing C–N bond gave rise to the highly atroposelective synthesis of C–N axially chiral 3-arylquinazolinones **193**. Comparing peptides catalyst **Cat-26** to **Cat-29**, it was demonstrated that **Cat-26**, when substituted with aminocyclopropane carboxylic acid (Acpc) on the labeled position, possessed the highest catalytic activation. Moreover, the terminal tertiary amine was essential for this enantioselective process due to the enhanced hydrogen bond acceptor ability (for details, see **Int-56**). Finally, further elaborations of both a dehalogenation Suzuki–Miyaura cross-coupling sequence and a regioselective Buchwald–Hartwig amination fabricated the products **193a–g** to obtain the complexes **194** and **195**, respectively.

Thiazines are a valuable class of heterocyclic scaffolds that have been widely utilized in pharmaceuticals and agrochemicals. Very recently, Jin et al. disclosed an asymmetric cyclization for the construction of thiazine derivatives **198** containing a C–N axis (Figure 52) [131]. The keys to achievements in this process involved both a NHC-catalyzed addition of thiourea to ynal-derived acetylenic acylazolium species and the subsequent intramolecular lactamation, finally leading to approaching atropisomeric thiazines **198** with excellent enantioselectivities. Moreover, the Lewis acid Sc(OTf)_3_ was essential for the catalytic transformation in both the enhancement of the reaction yield and retention of the enantioselectivity. Finally, this optical pure compound was easy to apply to the halogenation reactions, giving access to the chloro-, bromo-, and alkynyl-substituted products **199**, **200**, and **202**.

### 3.5. Synthetic Strategies to Access Miscellaneous Cyclic C–N Axially Chiral N-Arylamides

Ynamides are practical and valuable synthons employed in a variety of synthetic methodologies. In 2007, Hsung and co-workers disclosed a rhodium(I)-catalyzed enantioselective [2 + 2 + 2] annulation of 1,6-diynes **11** with achiral ynamides **203**, yielding enantiomerically enriched N,O-biaryls **204** and **205**, bearing both the C–C and C–N axis via the intermediate **Int-55** (Figure 53) [132]. This transformation offered a concise and useful synthetic method to access chiral N,O-biaryls and chiral anilides.

The development of an efficient methodology to access medium-sized *N*-heterocycles remains a challenging goal in synthetic organic chemistry, due to the unfavorable entropy effect and transannular interactions. In 2016, Enders and coworkers reported an NHC-catalyzed enantioselective, formal [3 + 4] annulation of isatin-derived enals **206** with *N*-(*ortho*-chloromethyl)arylamides **207** to construct a variety of spirobenzazepinones **208** bearing a quaternary all-carbon stereocenter and a C–N axial axis, offering good yields with excellent *ee* values (Figure 54) [133]. By this route, the nucleophilic addition of activated NHC **Cat-31** to the enal **206** formed a mesomeric azolium homoenolate, **Int-56.** A subsequent attack of **Int-56** to the in situ-generated *aza*-*o*-quinonemethide **209** from the substrate **207** led to access the key intermediate acyl azolium **Int-57**, followed by an intramolecular cyclization to afford the desired product **208**. Subsequent reduction of the amide and tosyl fragments of product **208a** led to the development of the sprio-compounds **210a** and **210b**, with the complete retention of the enantiopurity, respectively.

## 4. Conclusions and Perspectives

In summary, manifold catalytic atroposelective methodologies have been successfully achieved for the construction of atropisomeric C–N skeletons possessing diverse structures, such as acyclic/cyclic C–N axially chiral amides. Although rapid developments have been seen in this field, the methods for the enantioselective synthesis of C–N axially chirality are deficient, compared to the well-established strategies for biaryl atropisomers. Future efforts and advances need to be focused on the establishment of more robust, useful, general, concise, and efficient original catalytic asymmetric systems to synthesize these elegant entities. Furthermore, the merging of enabling and advanced techniques, such as flow chemistry, photochemistry, mechanochemistry, and electrochemistry, with the atroposelective formation of C–N atropisomers would be highly desired and sought after. Last but not least, the employment of these optically pure axially chiral C–N units as novel types of chiral ligands, catalysts, and pharmaceuticals should be extensively conducted.

## Data Availability

Not applicable.

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
