# Peer review of "Construction of Non-Biaryl Atropisomeric Amide Scaffolds Bearing a C–N Axis via Enantioselective Catalysis"

_molecules, 2022, doi:10.3390/molecules27196583_

Round 1
Reviewer 1 Report
With the rapid development of methodology for asymmetric atropisomeric compounds over the past decades, stereoselective construction of atropisomers, such as atropisomeric C-N compounds, has emerged as a hot topic and has attracted considerable attention. As a consequence, significant achievement has been made in this field. Xiao, Chen and co-workers in this manuscript elegantly summarized the development in this emerging field and gave some insights into future advance.
This manuscript will be greatly helpful in attracting related researchers to join this flourishing and popular research field, and also would inspire some new reaction of C-N atropisomers construction in the near future. In light of this, this reviewer recommends to publish it on the Molecules after addressing following issues:
1. Scheme 1a and Scheme 1b has not been displayed in the Scheme 1. Please revise it.
2. In scheme 17, the “Synthetic applications” should be revised to “Synthetic application”.
3. In the last paragraph of page 2, the “strategy” should be revised to “strategies”.
4. In the scheme 20 of page 17, “5A” should be revised to “5Å”.
5. In the scheme 21 of page 18 and scheme 30 of page 26, the letters of “dr” should be italic.
6. In the scheme 3 of page 4, The letters of “N” in Allylic N-Alkylation should be italic.
7. In the scheme 9 of page 8, The letters of “N” in N-Aryl Peptoid should be italic.
8. In the last paragraph of page 9, “sp3” should be revised to “sp3”.
9. In the scheme 45 of page 38, the letters of “ee” should be italic.
10. The length of bond in “C-H” should be uniform in all the schemes.
Author Response
Q1: Scheme 1a and Scheme 1b have not been displayed in the Scheme 1. Please revise it.
A1: Thanks for your suggestion. Scheme 1a and Scheme 1b have been displayed in the revised Scheme 1.
Q2: In scheme 17, the “Synthetic applications” should be revised to “Synthetic application”.
A2: Thanks for your suggestion. The “Synthetic applications” has been revised to “Synthetic application”.
Q3: In the last paragraph of page 2, the “strategy” should be revised to “strategies”.
A3: Thanks for your suggestion. The word of “strategy” should be revised to “strategies”. Additionally, this sentence has been revise from “In this Minireview, we aim to summarize breakthrough and recent advances in asymmetric catalyzed functionalization for construction of C–N axially chiral amide scaffold enabled by both metal catalysis and organocatalysis strategy.” to “In this minireview, we aim to summarize breakthrough and recent advances in asymmetric catalyzed functionalizations for the construction of C–N axially chiral amide scaffolds enabled by both metal catalysis and organocatalysis strategies.
”
Q4: In the scheme 20 of page 17, “5A” should be revised to “5Å”.
A4: Thanks for your suggestion. “5A” has been revised to “5Å”. Additionally, Scheme 20 has been revised to scheme 19.
Q5: In the scheme 21 of page 18 and scheme 30 of page 26, the letters of “dr” should be italic.
A5: Thanks for your significant suggestions. the letters of “dr” has been revised in the scheme 21 of page 18 and scheme 30 of page 26. Additionally, Scheme 21 and scheme 30 have been revised to scheme 20 and scheme 29, respectively.
Q6: In the scheme 3 of page 4, The letters of “N” in Allylic N-Alkylation should be italic.
A6: Thanks for your suggestion. The letters of “N” in Allylic N-Alkylation has be revised. Additionally, Scheme 3 has been revised to scheme 2.
Q7: In the scheme 9 of page 8, The letters of “N” in N-Aryl Peptoid should be italic.
Q7: Thanks for your suggestion. The letters of “N” in Allylic N-Alkylation has be revised. Additionally, the same mistakes have been corrected. Additionally, Scheme 9 has been revised to scheme 8.
Q8: In the last paragraph of page 9, “sp3” should be revised to “sp3”.
A8: Thanks for your significant suggestion. “sp3” should be revised to “sp3” in the last paragraph of page 9. Additionally, the same mistakes have been revised in the tile of scheme 10 and ref 53.
Q9: In the scheme 45 of page 38, the letters of “ee” should be italic.
A9: Thanks for your significant suggestion. the letters of “ee” has been revised. Additionally, Scheme 45 has been revised to scheme 44.
Q10: The length of bond in “C-H” should be uniform in all the schemes.
A10: Thanks for your significant suggestion. The length of bond in “C-H” has been revised to “C−H” in the manuscript.

Reviewer 2 Report
In this paper, the authors reports a critical review regarding the construction of non-biaryl atropisomeric amide scaffolds bearing a C–N axis via enantioselective catalysis. Overall, the paper describes interesting transformations and reaction concepts of an important current area of interest that from a synthetic point of view might be very useful in organic chemistry. So, I recommend the content of this manuscript to be published in Molecules.
Author Response
Q1: English language and style are fine/minor spell check required.
A1: Thanks for your significant suggestion. English language and style have been revised in this manuscript. Details are as follows:
- In line of 37, the sentence of “To date, the C-N axially chiral unit presented a series of significant natural products and bioactive molecules, which usually exhibit different pharmacological activities and metabolic processes in vivo and in vitro (Scheme 1a).” has been corrected to “To date, the C-N axially chiral units have presented a series of significant natural products and bioactive molecules, which usually exhibit different pharmacological activities and metabolic processes in vivo and in vitro (Scheme 1a).”
- In line of 61-63, This sentence of “In this Minireview, we aim to summarize breakthrough and recent advances in asymmetric catalyzed functionalization for construction of C–N axially chiral amide scaffold enabled by both metal catalysis and organocatalysis strategy.” has been corrected to “In this minireview, we aim to summarize breakthrough and recent advances in asymmetric catalyzed functionalizations for the construction of C–N axially chiral amide scaffolds enabled by both metal catalysis and organocatalysis strategies.
- In line of 65-67, the sentence of “For clarification, this minireview was organized into two sections according to the generation of different types of chiral products, including acyclic C–N axially chiral amide and cyclic C–N axially chiral amide.” has been corrected to “For clarification, this minireview is organized into two sections according to the generation of different types of chiral products, including acyclic C–N axially chiral amides and cyclic C–N axially chiral amides.”
- In line of 133-134, the sentence of “Reductive elimination of rhodium species releases the rhodium catalyst and formed the desired product (S)-12.” has been corrected to “Reductive elimination of rhodium species releaseed the rhodium catalyst and formed the desired product (S)-12.”
- In line of 205, the phrase of “N-methyl atropisomeric thioanilides (R)-28 was readily obtained” has been corrected to “N-methyl atropisomeric thioanilides (R)-28 were readily obtained”
- In line of 339-341, the sentence of “On a similar note, isothiourea-catalyzed atroposelective N‑acylation of sulfonamides by employing the ent-Cat-6 was delineated by Zhao and co-workers (Scheme 18).” has been corrected to “On a similar note, an isothiourea-catalyzed atroposelective N‑acylation of sulfonamides by employing the ent-Cat-6 was delineated by Zhao and co-workers (Scheme 18).”
- In line of 436-439, the sentence of “This section covers the most pertinent examples of synthesizing acyclic C–N axially chiral amide from prochiral precursors, such as the arylmaleimide, atropisomeric urazole, spirobenzazepinone, arylpyridones, arylquinazolinones, and the other types of atropisomeric anilide.” has been corrected to “This section covers the most pertinent examples of synthesizing acyclic C–N axially chiral amides from prochiral precursors, such as the arylmaleimides, atropisomeric urazoles, spirobenzazepinones, arylpyridones, arylquinazolinones, and the other types of atropisomeric anilides.”
- In line of 592-593, the sentence of “In this transformation, two atropisomers 131 and 132 was formed in high yields.” has been corrected to “In this transformation, two atropisomers 131 and 132 were formed in high yields.”
- In line of 752-7533, the phrase of “Selected C(sp2)-Br bond cleavage and gram-scale operation was smoothly carried out,” has been corrected to “Selected C(sp2)-Br bond cleavage and gram-scale operation were smoothly carried out,”

Reviewer 3 Report
Reviewers’ comments for the Manuscript ID: Molecules-1918824
The manuscript title: “Construction of Non-Biaryl Atropisomeric Amide Scaffolds Bearing a C–N Axis via Enantioselective Catalysis “by Xiao Xiao, et al. in the current review article authors comprehensively summarized various synthetic approached reported in the literature for construction of non -biaryl atropisomers rotating around a C-N chiral axis. Recently, Yong-JieWu et al also summarized the various methods for synthesis of atropisomers featuring a C–N chiral axis and published in Green Synthesis and Catalysis 3 (2022) 117–136”. But substrate scope and adequate literature evidence were not provided, but in the current paper, authors collected almost all available literature and clearly described about them, this information will be very useful for future research endeavors, so, I would strongly, recommend this review article for publication in Molecules journal in its current form. However, below are few general comments should be addressed before publication.
General comments
1) Authors should cite “Green Synthesis and Catalysis 3 (2022) 117–136” paper
2) In line 21 authors used acid catalysis reaction, et al., there is no meaning for et al in this line and throughout the manuscripts, so authors should change it etc.
3) In the first part authors mentioned scheme 1a and scheme 1b, but there is no scheme 1a and 1b, it should be figure 1a and 1b.
4) It would be better giving description for R1-R5 in scheme-2 and scheme-2 should be changed to Figure2
5) In line 86 authors mentioned allyl acetate 2, it should be corrected to Diallyl carbonate
6) in scheme -3 uniform numbering missing for structure 3a-g and describe R2 in each structure
Author Response
Q1: Authors should cite “Green Synthesis and Catalysis 3 (2022) 117–136” paper.
A1: Thanks for your significant suggestion. This review paper was cited in the original manuscript (ref 20).
Q2: In line 21 authors used acid catalysis reaction, et al., there is no meaning for et al in this line and throughout the manuscripts, so authors should change it etc.
A2: Thanks for your significant suggestion. The sentence of “…acid catalysis reaction, et al.” has been revised to “…acid catalysis reaction, etc.” The same mistakes have been corrected in this article.
Q3: In the first part authors mentioned scheme 1a and scheme 1b, but there is no scheme 1a and 1b, it should be figure 1a and 1b.
A3: Thanks for your suggestion. Scheme 1a and Scheme 1b have been displayed in the revised Scheme 1.
Q4: It would be better giving description for R1-R5 in scheme-2 and scheme-2 should be changed to Figure2.
A4: Thanks for your suggestion. R1-R5 in scheme-2 have been given description in the revised manuscript. Moreover, scheme-2 has been changed to Figure 1.
Q5: In line 86 authors mentioned allyl acetate 2, it should be corrected to Diallyl carbonate.
A5: Thanks for your suggestion. The mentioned allyl acetate 2 has been revised to diallyl carbonate.
Q6: in scheme -3 uniform numbering missing for structure 3a-g and describe R2 in each structure
A6: Thanks for your suggestion. The scheme 3 has been changed to scheme 2 in the revised manuscript now. All the mistakes you mentioned in scheme 2 has been corrected in the revised manuscript.

Reviewer 4 Report
1. The work is more of a monograph, a string of syntheses from several articles. It is rather advisable to publish a book chapter.
2. I don't see any connection with the domain of the magazine.
3. None of the written reactions are verified experimentally through various analyses.
4. It is only a theoretical work, without a characterization of the obtained products.
5. With what yield are the syntheses obtained? What is the proportion of secondary phases?
6. The reactions are written very small and are not clearly visible. They should be enlarged and rewritten with editing programs.
7. I don't see any citations for the 55 schemes. There should be approval from the authors initially that they allow their use in the paper.
8. I recommend publication as a book chapter, the work does not have the quality of a review.
Author Response
Q1: The work is more of a monograph, a string of syntheses from several articles. It is rather advisable to publish a book chapter.
A1: Thanks for your suggestion. This review paper is submitted for the special issue "Atroposelective Synthesis of Novel Axially Chiral Molecules". We have organized this review into two sections according to the generation of different types of chiral products, including acyclic C–N axially chiral amide and cyclic C–N axially chiral amide. Moreover, the analyses of synthetic merits and disadvantages for each example have been displayed as well.
Q2: I don't see any connection with the domain of the magazine.
A2: Thanks for your suggestion. This review paper is submitted for the special issue "Atroposelective Synthesis of Novel Axially Chiral Molecules"
Q3: None of the written reactions are verified experimentally through various analyses.
A3: Thanks for your suggestion. The analyses of synthetic merits and disadvantages for each example have been displayed as well.
Q4: It is only a theoretical work, without a characterization of the obtained products.
A4: Thanks for your suggestion. In this review, we have not characterized all of the obtained products, but the represented products, such as the products in scheme 4, scheme 6, scheme 12, and scheme 20, have been described in this review.
Q5: With what yield are the syntheses obtained? What is the proportion of secondary phases?
A5: Thanks for your suggestions. The yield of all the products presented in the review have given.
Q6: The reactions are written very small and are not clearly visible. They should be enlarged and rewritten with editing programs.
A6: Thanks for your suggestions. All of the reaction pictures have been enlarged in the revised manuscript. For details, please see the figure and schemes in the revised manuscript.
Q7: I don't see any citations for the 55 schemes. There should be approval from the authors initially that they allow their use in the paper.
A7: Thanks for your suggestions. All of the schemes and figure are not copied from the reported papers, which are drawn by us with our own understanding. Furthermore, we have cited the reference followed the related schemes.
Q8: I recommend publication as a book chapter, the work does not have the quality of a review.
A8: Thanks for your suggestion. The analyses of synthetic merits and disadvantages for each example have been displayed as well, thereby the work has the quality of a review. Additionally, we have corrected some mistakes of English language and style in the revised manuscript. Details are as follows:
- In line of 37, the sentence of “To date, the C-N axially chiral unit presented a series of significant natural products and bioactive molecules, which usually exhibit different pharmacological activities and metabolic processes in vivo and in vitro (Scheme 1a).” has been corrected to “To date, the C-N axially chiral units have presented a series of significant natural products and bioactive molecules, which usually exhibit different pharmacological activities and metabolic processes in vivo and in vitro (Scheme 1a).”
- In line of 61-63, This sentence of “In this Minireview, we aim to summarize breakthrough and recent advances in asymmetric catalyzed functionalization for construction of C–N axially chiral amide scaffold enabled by both metal catalysis and organocatalysis strategy.” has been corrected to “In this minireview, we aim to summarize breakthrough and recent advances in asymmetric catalyzed functionalizations for the construction of C–N axially chiral amide scaffolds enabled by both metal catalysis and organocatalysis strategies.
- In line of 65-67, the sentence of “For clarification, this minireview was organized into two sections according to the generation of different types of chiral products, including acyclic C–N axially chiral amide and cyclic C–N axially chiral amide.” has been corrected to “For clarification, this minireview is organized into two sections according to the generation of different types of chiral products, including acyclic C–N axially chiral amides and cyclic C–N axially chiral amides.”
- In line of 133-134, the sentence of “Reductive elimination of rhodium species releases the rhodium catalyst and formed the desired product (S)-12.” has been corrected to “Reductive elimination of rhodium species releaseed the rhodium catalyst and formed the desired product (S)-12.”
- In line of 205, the phrase of “N-methyl atropisomeric thioanilides (R)-28 was readily obtained” has been corrected to “N-methyl atropisomeric thioanilides (R)-28 were readily obtained”
- In line of 339-341, the sentence of “On a similar note, isothiourea-catalyzed atroposelective N‑acylation of sulfonamides by employing the ent-Cat-6 was delineated by Zhao and co-workers (Scheme 18).” has been corrected to “On a similar note, an isothiourea-catalyzed atroposelective N‑acylation of sulfonamides by employing the ent-Cat-6 was delineated by Zhao and co-workers (Scheme 18).”
- In line of 436-439, the sentence of “This section covers the most pertinent examples of synthesizing acyclic C–N axially chiral amide from prochiral precursors, such as the arylmaleimide, atropisomeric urazole, spirobenzazepinone, arylpyridones, arylquinazolinones, and the other types of atropisomeric anilide.” has been corrected to “This section covers the most pertinent examples of synthesizing acyclic C–N axially chiral amides from prochiral precursors, such as the arylmaleimides, atropisomeric urazoles, spirobenzazepinones, arylpyridones, arylquinazolinones, and the other types of atropisomeric anilides.”
- In line of 592-593, the sentence of “In this transformation, two atropisomers 131 and 132 was formed in high yields.” has been corrected to “In this transformation, two atropisomers 131 and 132 were formed in high yields.”
- In line of 752-7533, the phrase of “Selected C(sp2)-Br bond cleavage and gram-scale operation was smoothly carried out,” has been corrected to “Selected C(sp2)-Br bond cleavage and gram-scale operation were smoothly carried out,”

Round 2
Reviewer 4 Report
1. It is a complex study but I have some objections to the structuring of the manuscript. At the end of the introduction, the authors described the purpose (objectives) of the study, mentioning also the methods used, obviously briefly, but this should be explained in more detail in the methods. In terms of materials and methods, each technique applied should appear as a subpoint of this chapter, so it would be easier to follow the results as well. In the conclusions appear aspects that should appear in the chapter of results and be described only strictly the conclusions of this study. Do you consider that the topic is relevant in the research area and if so, what is it? I believe that the authors should focus more on the usefulness of these materials and, depending on their structural and magnetic properties. In case of conclusions, you could explain what does it add to the subject area compared to other published material?
2. Abstract: Describe more relevant results in the abstract. Mention the purpose for which this study was conducted.
4. The lengthy sentences may be split in to smaller sentence without change of its meaning.
5. Also, suggested to include the recent references in the introduction part.
6. The results and discussions part should be compared with the literature data. To redo the part of results and discussions by a systematic presentation of the results by which the readers of the articles manage to follow the article more easily..
7. Figure quality is poor throughout. To improve the quality of the figures. Enlarge the characters in the figures.
8. To correlate the results obtained with the results present in other works.
9. It is a complex study but I have some objections to the structuring of the manuscript. At the end of the introduction, the authors described the purpose (objectives) of the study, mentioning also the methods used, obviously briefly, but this should be explained in more detail in the methods. In terms of materials and methods, each technique applied should appear as a subpoint of this chapter, so it would be easier to follow the results as well. In the conclusions appear aspects that should appear in the chapter of results and be described only strictly the conclusions of this study. Do you consider that the topic is relevant in the research area and if so, what is it? In case of conclusions, you could explain what does it add to the subject area compared to other published material?
10. Conclusions should be short with important observations.
11. References are not written in unison. Some journals are abbreviated and some are not.
12, To pass the work in the materials page. Neither the format of the diary nor the size of the figures with the related legends are respected.
Author Response
Q1: 1. It is a complex study but I have some objections to the structuring of the manuscript. At the end of the introduction, the authors described the purpose (objectives) of the study, mentioning also the methods used, obviously briefly, but this should be explained in more detail in the methods. In terms of materials and methods, each technique applied should appear as a subpoint of this chapter, so it would be easier to follow the results as well. In the conclusions appear aspects that should appear in the chapter of results and be described only strictly the conclusions of this study. Do you consider that the topic is relevant in the research area and if so, what is it? I believe that the authors should focus more on the usefulness of these materials and, depending on their structural and magnetic properties. In case of conclusions, you could explain what does it add to the subject area compared to other published material?
Q1-1: It is a complex study but I have some objections to the structuring of the manuscript. At the end of the introduction, the authors described the purpose (objectives) of the study, mentioning also the methods used, obviously briefly, but this should be explained in more detail in the methods. In terms of materials and methods, each technique applied should appear as a subpoint of this chapter, so it would be easier to follow the results as well.
A1-1: Thanks for your suggestion. In this review, we want to present a fully clarification, including reaction method, mechanism, late-stage transformation, and application, to the readers. If they are interested in the detailed method and technique applied, the original work is probably the best choice. Therefore, the structuring of our manuscript may be the preferable format.
Q1-2: In the conclusions appear aspects that should appear in the chapter of results and be described only strictly the conclusions of this study. Do you consider that the topic is relevant in the research area and if so, what is it? I believe that the authors should focus more on the usefulness of these materials and, depending on their structural and magnetic properties. In case of conclusions, you could explain what does it add to the subject area compared to other published material?
A1-2: Thanks for your suggestion. The topic of this special issue is “Atroposelective Synthesis of Novel Axially Chiral Molecules”. Our manuscript was structured around the topic of atroposelective construction of non-biaryl atropisomeric amide scaffolds. Compared to other published material, this manuscript elaborately introduced the catalytic methodology to access non-biaryl atropisomeric amide scaffolds.
Q2: 2. Abstract: Describe more relevant results in the abstract. Mention the purpose for which this study was conducted.
A2: Thanks for your suggestion. The purpose of this study has been described in the revised manuscript. The sentence of “In this review, we summarize the development of catalytic asymmetric synthetic strategies to access non-biaryl atropisomers rotating around a C‒N chiral axis.” has been revised to “In this review, we summarize the development of catalytic asymmetric synthetic strategies to access non-biaryl atropisomers rotating around a C‒N chiral axis, including reaction methods, mechanism, late-stage transformations, and applications.”
Q3: 4. The lengthy sentences may be split in to smaller sentence without change of its meaning.
A3: Thanks for your suggestion. The lengthy sentences have been split into smaller sentence without change of its meaning. For details, please see as follows:
- In Page 7, the sentence has been revised from “To fill this gap, in 2020, Shi and co-workers described the first formal Pd(II)-catalyzed atroposelective C−H olefination to access a highly efficient synthesis of atropisomeric anilides 15 via using readily available L-pyroglutamic acid (L5) as the chiral ligand, which underwent both an asymmetric C−H functionalization and a dynamic kinetic resolution process (Scheme 6).[50]” to “To fill this gap, in 2020, Shi and co-workers described the first formal Pd(II)-catalyzed atroposelective C−H olefination to access a highly efficient synthesis of atropisomeric anilides 15 via using readily available L-pyroglutamic acid (L5) as the chiral ligand (Scheme 6).[50] This reaction underwent both an asymmetric C−H functionalization and a dynamic kinetic resolution process.”
- In Page 7, the sentence has been revised from “Experimental studies of the N−Ar rotational barriers (ΔG‡) and a negative correlation between δCO and the rotational barriers elucidated that the atropostability of those atropisomeric anilides toward racemization depended on the electronic effects of both aromatic ring and the picolinamide.” to “Experimental studies were conducted to investigate the N−Ar rotational barriers (ΔG‡), and a negative correlation between δCO and the rotational barriers. These elucidated that the atropostability of those atropisomeric anilides toward racemization depended on the electronic effects of both aromatic ring and the picolinamide.”
- In Page 9, the sentence has been revised from “In this conversion, pyridine was indispensable which served as the coordination site to accelerate the C–H activation rather than the carbonyl group or peptoid backbone, and the steric hindrance on pyridine group would inhibit the coordination ability of palladium.” to “In this conversion, pyridine was indispensable which served as the coordination site to accelerate the C–H activation rather than the carbonyl group or peptoid backbone. Moreover, the steric hindrance on pyridine group would inhibit the coordination ability of palladium.”
- In Page 42, the sentence has been revised from “Mechanistically, the reaction began with an oxidization to access CpRhIII from the chiral CpRhI by using AgNTf2 as oxidant, followed by a coordination of the oxygen atom of 152a to the RhIII catalyst promoting the first C−H bond activation via a concerted metalation-deprotonation (CMD) mechanism, and leading to the formation of the six-membered rhodacyclic intermediate Int-36.” to “Mechanistically, the reaction began with an oxidization to access CpRhIII from the chiral CpRhI by using AgNTf2 as oxidant. A subsequent coordination of the oxygen atom of 152a to the RhIII catalyst promoting the first C−H bond activation occurred through a concerted metalation-deprotonation (CMD) mechanism, leading to the formation of the six-membered rhodacyclic intermediate Int-36.”
Q4: 5. Also, suggested to include the recent references in the introduction part.
A4: Thanks for your suggestion. The recent references (ref 5, and ref. 6) have been added in the introduction part.
Q5: 6. The results and discussions part should be compared with the literature data. To redo the part of results and discussions by a systematic presentation of the results by which the readers of the articles manage to follow the article more easily.
A5: Thanks for your suggestion. All the results of literature data, such as yields, and ee and dr values, have been present in the corresponding schemes, and the readers can obtain the results from the scheme easily. Additionally, the omissive results of literature data in the scheme 33, scheme 38, scheme 52, and scheme 54.
Q6: 7. Figure quality is poor throughout. To improve the quality of the figures. Enlarge the characters in the figures.
A6: Thanks for your suggestion. The characters in all figures have been enlarged.
Q7: 8. To correlate the results obtained with the results present in other works.
A7: Thanks for your suggestion. This minireview is organized into two sections according to the generation of different types of chiral products, including acyclic C–N axially chiral amide and cyclic C–N axially chiral amide. So, all the results have the correlations.
Q8: 9. It is a complex study but I have some objections to the structuring of the manuscript. At the end of the introduction, the authors described the purpose (objectives) of the study, mentioning also the methods used, obviously briefly, but this should be explained in more detail in the methods. In terms of materials and methods, each technique applied should appear as a subpoint of this chapter, so it would be easier to follow the results as well. In the conclusions appear aspects that should appear in the chapter of results and be described only strictly the conclusions of this study. Do you consider that the topic is relevant in the research area and if so, what is it? In case of conclusions, you could explain what does it add to the subject area compared to other published material?
A8: Thanks for your suggestion. This query you pose is repeated with the Q1.
Q9: 10. Conclusions should be short with important observations.
A9: Thanks for your suggestion. In the conclusion, the last two sentences have been revised from “Moreover, the employment of these optically pure axially chiral C–N units as the novel type of chiral ligands, catalysts and pharmaceuticals should be extensively conducted. Last but not least, the other rare axially chiral structures (such as C–O, C–B or N–N axis) should also be synthesized by exploring efficient methodologies.[19]” to “Last but not least, the employment of these optically pure axially chiral C–N units as the novel type of chiral ligands, catalysts and pharmaceuticals should be extensively conducted.”
Q10: 11. References are not written in unison. Some journals are abbreviated and some are not.
A10: Thanks for your suggestion. We have checked and revised all the reference. Most of journals are abbreviated except the journals of Nature, Science, Synthesis, Tetrahedron, Synlett, and Heterocycles.
Q11: 12, To pass the work in the materials page. Neither the format of the diary nor the size of the figures with the related legends are respected.
A11: Thanks for your suggestion. It’s difficult to revise our manuscript as you mentioned now.
